# THE ETHICS OF PRIVACY-PRESERVING DEEP LEARNING: THE GOOD, THE BAD, AND THE UGLY

## ABSTRACT

Homomorphic Encryption (HE) is gaining traction in Artificial Intelligence (AI) as a solution to privacy concerns, particularly in sensitive areas such as health-care. HE enables computation directly on encrypted data without ever decrypting it, ensuring that private information remains protected throughout the process, effectively enabling Privacy-Preserving Machine and Deep Learning (PP-MDL). While much of the discussion focuses on its privacy benefits, little attention has been paid to the ethical implications of using and training AI models on encrypted data, especially when the resulting models themselves remain encrypted and opaque to both developers and users. In this paper, we explore three ethical perspectives on the use of HE for PP-MDL: the Good, i.e., the clear advantages in terms of privacy and data protection; the Bad, i.e., the practical and conceptual ethical challenges it introduces; and the Ugly, i.e., the subtle, unexpected ethical issues that may arise when HE-powered PP-MDL is deployed in the real-world. Our aim is to show that while HE can strengthen privacy, it is not a silver bullet for ethical AI. It can complicate accountability, transparency, and trust, raising important ethical and societal questions that should not be overlooked.

## 1 INTRODUCTION

Homomorphic Encryption (HE) is increasingly recognized as a powerful tool for enabling privacy-preserving Artificial Intelligence (AI) applications Campbell (2022), in particular involving Machine and Deep Learning (ML and DL). By allowing computations directly on encrypted data, HE offers a compelling solution to one of the core challenges in modern data-driven systems: how to harness the power of AI without compromising sensitive personal information, effectively enabling privacy-preserving ML and DL (PP-MDL). This is particularly relevant in domains such as health-care, where legal protections are stringent Munjal & Bhatia (2023).

Yet, while the technical appeal of HE is clear, its ethical consequences remain underexplored. Most discussions emphasize the cryptographic strength and privacy benefits of HE Campbell (2022), often underpinned by an implicit assumption that stronger data confidentiality inherently implies ethical soundness, an assumption that warrants critical scrutiny in real-world AI deployments. Indeed, the implications of the use of HE in AI on ethical values like accountability, transparency, responsibility, among the others, are not sufficiently explored. The same holds for the explicit or implicit trade-offs introduced by HE. This paper aims to fill this gap in the literature, by providing an analysis of the direct and indirect ethical consequences of the use of HE in AI services.

To frame the ethical discussion, after the background (Sec. 3) and related literature (Sec. 2), we first formalize PP-MDL services under HE (Sec. 4). This is necessary because key aspects like encrypted training, hidden loss functions, and the impossibility of human-in-the-loop validation, are often unintuitive. Establishing this foundation helps make the ethical analysis clearer and more grounded. We then turn to The Good (Sec. 5), showing how HE can support ethically valuable goals, such as privacy compliance. In The Bad (Sec. 6), we explore trade-offs that arise directly from HE's properties, especially around responsibility, transparency, and accountability. Finally, The Ugly (Sec. 7) focuses on second-order issues (i.e., broader ethical and societal implications) that surface when PP-MDL systems are deployed in the real-world, where multiple actors, legal frameworks, and ethical expectations intersect.

We structure our inquiry around three guiding research questions:

RQ1 (The Good): How does HE contribute to ethically desirable outcomes in the design and deployment of PP-MDL systems?

RQ2 (The Bad): What direct ethical trade-offs arise from the use of HE in PP-MDL, particularly concerning accountability, interpretability, and responsibility?

RQ3 (The Ugly): In what ways might HE be used to mask or enable unethical AI practices in the real-world, despite its privacy-preserving aims?

This paper shifts the focus from the technical advances of HE to the ethical challenges it raises when enabling PP-MDL services. While HE research has pushed cryptographic and algorithmic frontiers, systematic ethical reflection on its real-world risks is still missing and urgently needed. By adopting an interdisciplinary lens, this work explores technical issues through an ethical perspective, aiming both to address some challenges by design and to foster greater awareness, disproving the claim that sees HE as a free pass for ethical AI.

## 2 RELATED LITERATURE

In recent years, the literature on PP-MDL with HE has expanded, thanks to the advancements in the mathematical and technological foundations of HE. It is noteworthy that, as of today, the large majority of works in this field focus on inference, given that the drawbacks of HE severely limit the possibility of encrypted training.

The first notable example of applications of HE to PP-MDL systems is CryptoNets Gilad-Bachrach et al. (2016), which demonstrated the feasibility of encrypted inference using Convolutional Neural Networks on encrypted images. After that, several works have explored optimizations in this context (Hesamifard et al. (2017); Boemer et al. (2019); Lou et al. (2020)). Other interesting applications of encrypted inference include anomaly detection Falcetta & Roveri (2024), chatbots Rovida & Leporati (2024), and time-series processing Falcetta & Roveri (2022). Notably, examples of applications in the medical domain are appearing, such as genotype imputation Gürsoy et al. (2022) and cancer classification Colombo et al. (2024). For what concerns the encrypted training, the first works focused on simple models such as logistic regression Kim et al. (2018), often providing limited security guarantees (e.g., 80-bits of security level). More recently, few works propose the training of DL models using backpropagation, under several constraints and conditions (e.g., Yoo & Yoon (2021); Colombo et al. (2025)).

The ethical discussion around HE is limited, but gaining attention. Ahmed & Hrzic (2025) explores the ethical aspects of hybrid systems integrating HE and blockchain for genomic and health data. Instead, Scheibner et al. (2022) conducted a study, by interviews, with health experts in Switzerland in order to understand how HE is perceived by medical experts, with a focus on ethical issues such as informed consent. Falcetta et al. (2023) explored the ethical trade-offs in personalizing AI solutions for health-care, also in the PP-MDL case. Lastly, Heinz et al. (2020) investigated the implications of the use of HE in the context of the European General Data Protection Regulation (GDPR).

## 3 BACKGROUND

HE is a class of encryption schemes that enable computations to be performed directly on encrypted data, without requiring decryption at any point in the process Boemer et al. (2019). Formally, given an encryption function $E(k_p, \cdot)$ and its corresponding decryption function $D(k_s, \cdot)$, the pair is said to be homomorphic with respect to a set of functions $\mathcal{F}$ if, for every $f \in \mathcal{F}$, there exists a function $g$ such that:

$$f(m) = D(k_s, g(E(k_p, m)))$$

for any input message $m$. Here, $k_p$ and $k_s$ denote the public and secret keys, respectively.

The core idea behind HE is that it preserves the algebraic structure of the underlying data: computations performed on ciphertexts yield encrypted results that, once decrypted, match the output of the same computation on the original plaintexts Ogburn et al. (2013). This property enables secure

computation outsourcing without compromising data confidentiality, i.e., the assurance that data is accessible only to those authorized to have access Stallings & Brown (2017).

HE schemes also uphold *semantic security* Brakerski et al. (2014), meaning that encrypting the same message multiple times yields distinct ciphertexts, thanks to injected randomness. Formally:

$$E(k_p, m) \neq E(k_p, m') \quad \text{even if} \quad m = m'.$$

This ensures that an attacker observing ciphertexts cannot deduce information about the original data or its processed form. From a functional perspective, HE schemes are categorized based on the types and number of operations they support. Partially HE supports a single operation, either addition or multiplication. Classical examples include Rivest et al. (1978) and Paillier (1999). Leveled HE supports bounded-depth arithmetic circuits, with parameters set in advance to limit noise accumulation. Examples include BGV Brakerski et al. (2014), BFV Fan & Vercauteren (2012), and CKKS Cheon et al. (2017). Lastly, Fully HE removes the operation limit entirely by leveraging a technique called *bootstrapping* Gentry (2009), which performs a ciphertext maintenance operation after the encrypted computation. An example is TFHE Chillotti et al. (2020), which focuses on fast binary gate evaluation.

The security of modern HE schemes relies on the hardness of the Ring Learning With Errors (RLWE) problem Lyubashevsky et al. (2013), a lattice-based assumption considered resistant to both classical and quantum attacks. RLWE extends the classic LWE problem to polynomial rings, enabling efficient encrypted computation while retaining strong theoretical guarantees through reductions to hard problems on ideal lattices.

Despite these advantages, the most pressing limitation of HE remains its computational cost. Operations on encrypted data, especially multiplications, are several orders of magnitude slower than their plaintext counterparts. Inference latency, ciphertext expansion, and the need for frequent noise management (e.g., bootstrapping) all contribute to the overhead. These costs are particularly pronounced in real-time applications or when using deep models with many layers and non-linearities. Currently, HE-based solutions still require careful approximation, model simplification, or hybridization to remain tractable in practice. However, hardware advances have progressively improved performance Jayashankar et al. (2025), and HE is expected to become practical for many real-world applications in the next few years Gorantala et al. (2023).

## 4 PROBLEM FORMULATION

In the following, we formalize a PP-MDL service based on HE, which will serve as the technical reference for the ethical analysis that follows. This abstraction is intentionally general and can be instantiated across diverse domains, such as image classification, anomaly detection, or text generation in chatbots. The aim is to define the core components and transformations that characterize a typical ML or DL service able to process encrypted data.

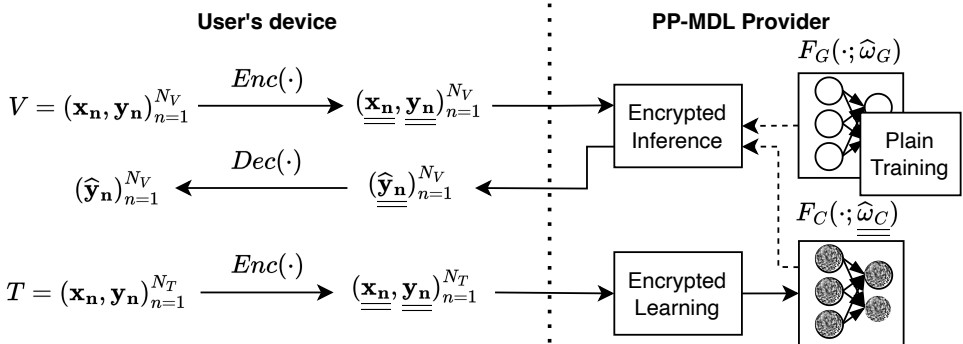

Figure 1: The two modalities of HE-based PP-MDL. In the PT-EI, the Plain Training module is used to train a generic model $F_G(\cdot; \widehat{\omega}_G)$, that is then used in the Encrypted Inference module. In the ET-EI modality, the Encrypted Learning module is used to train an encrypted custom model $F_C(\cdot; \underline{\widehat{\omega}}_C)$.

The presentation distinguishes between two primary usage modes. In the first, i.e., *Plain Training-Encrypted Inference* (PT-EI), a generic model $F_G(\cdot; \widehat{\omega}_G)$ is trained on plaintext data and subsequently used to perform inference on encrypted inputs. This setting offers strong data confidentiality for end-users while preserving training efficiency. In the second, i.e., *Encrypted Training-Encrypted Inference* (ET-EI), both training and inference occur on encrypted data, producing a custom encrypted model $F_C(\cdot; \widehat{\underline{\underline{\omega}}}_C)$, maximizing data confidentiality throughout the ML lifecycle. The two modes are shown in Fig. 1. Let:

$$D = \left\{ T = (\mathbf{x}_n, \mathbf{y}_n)_{n=1}^{N_T}, \quad V = (\mathbf{x}_n, \mathbf{y}_n)_{n=1}^{N_V} \right\} \tag{1}$$

be a dataset composed of a training set $T$ with $N_T$ samples and a test set $V$ with $N_V$ samples. Each sample contains a feature vector $\mathbf{x}_i \in \mathbb{R}^K$ and a target vector $\mathbf{y}_i \in \mathbb{R}^L$. Let:

$$F(\mathbf{x}, \omega) = \widehat{\mathbf{y}} \tag{2}$$

be a DL model parameterized by weights $\omega$, which maps inputs $\mathbf{x} \in \mathbb{R}^K$ to predictions $\widehat{\mathbf{y}} \in \mathbb{R}^L$.

In general, the learning objective is to estimate weights $\omega \in \Omega$ by minimizing a loss over the training set:

$$\mathcal{L}(\omega) = \frac{1}{N_T} \sum_{n=1}^{N_T} (\mathbf{y}_n, F(\mathbf{x}_n, \omega)), \tag{3}$$

$$\widehat{\omega} = \underset{\omega \in \Omega}{\arg\min} \, \mathcal{L}(\omega). \tag{4}$$

Once trained, the model $F(\cdot; \widehat{\omega})$ is used to make predictions on unseen samples from $V$.

## 4.1 PLAIN TRAINING-ENCRYPTED INFERENCE

In this modality, a generic model $F_G(\cdot; \widehat{\omega}_G)$ is trained on public plain data $T$ and later used to perform inference over encrypted inputs $\underline{\underline{\mathbf{x}}}$ coming from an user of the PP-MDL service. Input features are encrypted with the public key $\overline{\overline{k}}_p$:

$$\underline{\underline{\mathbf{x}_n}} = \text{Enc}(\mathbf{x}_n, k_p). \tag{5}$$

Predictions are computed homomorphically as:

$$\underline{\underline{\widehat{\mathbf{y}}_n}} = F_G(\underline{\underline{\mathbf{x}_n}}, \widehat{\omega}_G), \tag{6}$$

and decrypted by the user using the secret key $k_s$:

$$\widehat{\mathbf{y}}_n = \text{Dec}(\underline{\underline{\widehat{\mathbf{y}}_n}}, k_s). \tag{7}$$

This approach preserves input and output confidentiality and is currently the most practical and widely adopted HE-based solution.

## 4.2 ENCRYPTED TRAINING AND INFERENCE

In this modality, both the training set and inference inputs are encrypted. The encrypted training set is defined as:

$$\underline{\underline{T}} = \text{Enc}(T, k_p) = (\underline{\underline{\mathbf{x}_n}}, \underline{\underline{\mathbf{y}_n}})_{n=1}^{N_T}, \tag{8}$$

and the learning process becomes:

$$\underline{\underline{\mathcal{L}}}(\underline{\underline{\omega}}_C) = \frac{1}{N_T} \sum_{n=1}^{N_T} \left( \underline{\underline{\mathbf{y}_n}}, F_C(\underline{\underline{\mathbf{x}_n}}, \underline{\underline{\omega}}_C) \right), \tag{9}$$

$$\widehat{\underline{\underline{\omega}}}_C = \underset{\underline{\underline{\omega}}_C \in \underline{\underline{\Omega}}_C}{\arg\min} \, \underline{\underline{\mathcal{L}}}(\underline{\underline{\omega}}_C). \tag{10}$$

with the custom model being trained directly on encrypted data, obtaining encrypted weights $\widehat{\underline{\underline{\omega}}}_C$. This setting ensures data confidentiality also during the training, but introduces substantial computational and algorithmic challenges, including high latency, noise accumulation, and limitations in model expressivity due to HE constraints. While these issues remain the subject of active research and currently limit the practical deployment of fully encrypted training, for the purposes of this paper we assume both modalities, i.e., PT-EI and ET-EI, are viable. Our focus is not on overcoming these technical barriers, but on anticipating and analyzing the ethical implications that arise once such privacy-preserving AI services will become widely deployable.

## 5 THE GOOD

The primary ethical appeal of HE lies in its ability to protect data *in use*, not just at rest or in transit Campbell (2022). By ensuring that data remains encrypted throughout the entire computational pipeline, HE eliminates a broad class of risks related to accidental leaks, malicious access, or unauthorized processing. This level of protection is particularly valuable in domains governed by strict regulatory frameworks (such as health-care Meszaros et al. (2022), finance Boukherouaa et al. (2021), and defense Weng (2024)) where strong guarantees of data confidentiality are often legally mandated before any computation can take place. In such contexts, HE unlocks the possibility of performing analytics, training, or inference on sensitive data that would otherwise remain inaccessible due to legal or ethical constraints. This potential has significant implications for the ethical deployment of AI, both in terms of retrofitting existing services and enabling entirely new applications.

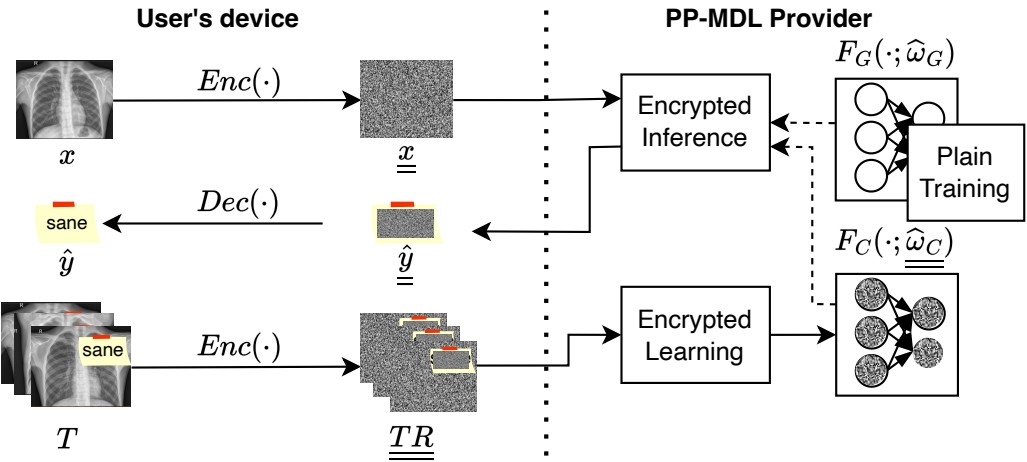

Figure 2: A diagnostic system based on medical images, such as X-rays, reimagined within the PP-MDL paradigm to enable both PT-EI and ET-EI.

HE allows many mainstream AI services to be re-engineered as privacy-preserving by design. Common use cases such as medical diagnosis from clinical data (Fig. 2), financial risk assessment, recommendation systems, or even large language model (LLM) based personal assistants can all benefit from encrypted inference. This is particularly true in the PT-EI modality, where models $F_G(\cdot; \widehat{\omega})$ are trained on plain public or controlled datasets $T$, but are deployed to make predictions on encrypted user inputs $\underline{x}$. For instance, HE can enable a hospital to access an AI-based diagnostic tool without disclosing patient records, or allow a user to query a recommendation system or chatbot without leaking their interests or health history. HE therefore acts as an ethical enabler for already existing AI services.

More profoundly, HE also makes it possible to conceive AI services that were previously unthinkable due to the severity of their privacy constraints. In highly sensitive domains (such as military intelligence, behavioral monitoring, neuroimaging, or genomics) any data sharing or centralization poses unacceptable risks. Thanks to the ET-EI modality, it is now possible to collaboratively train and deploy custom models $F_C(\cdot; \widehat{\underline{\omega}}_C)$ on such encrypted datasets $\underline{T}$ without ever revealing them. This shifts the design paradigm: instead of starting from functionality and retrofitting privacy, services can be designed from the beginning with privacy as a first-class citizen. For example, multiple hospitals across different jurisdictions may jointly train a disease progression model on encrypted patient data without violating local data protection laws. Such scenarios were previously highly complex from a technical standpoint and, moreover, not permitted under ethical and legal frameworks. With HE, they become possible.

Beyond protecting raw data, HE also safeguards the intent behind a computation, a critical but often overlooked aspect of privacy. In many real-world cases, the mere act of querying a model can

reveal sensitive contextual information. For instance, submitting a satellite image to detect military assets may expose a government's strategic interests, even if the image itself is not private. HE prevents the service provider from observing inputs, outputs, or the purpose of a task, offering a form of contextual privacy rarely achieved by other privacy-enhancing technologies. This aligns with Nissenbaum's theory of contextual integrity Nissenbaum (2004), which frames privacy as the protection of appropriate information flows within specific social contexts. In this view, privacy violations occur not simply from data exposure, but from contextual misuse, when information, or its use, deviates from its expected setting. By concealing not only the data but also its intended use, HE helps uphold these contextual boundaries, extending privacy from content to semantics.

Moreover, HE can serve as a foundational building block to enhance the security of other privacy-preserving techniques. For example, in Federated Learning, HE can be used to encrypt model updates exchanged between clients and the central server Jin et al. (2023), reducing the risk of information leakage from gradients or intermediate representations. HE also strengthens privacy-preserving search and retrieval mechanisms, such as Private Information Retrieval Song et al. (2023), by allowing encrypted queries over encrypted data. Finally, HE can be integrated with zero-knowledge proof systems to support verifiable computation, allowing one party to prove that a computation was correctly performed over encrypted data, without revealing the data or the computation itself Lee et al. (2025). In all these scenarios, HE complements existing approaches, contributing to a more robust and composable privacy-preserving ecosystem.

## 6 THE BAD

While HE provides strong privacy guarantees, it also introduces significant ethical trade-offs. Specifically, while it enables the promotion of privacy by allowing computation on encrypted data, it can simultaneously hinder other important values in AI, which may become more difficult to achieve under encryption. Many of these trade-offs stem directly from the opacity it enforces by design.

### 6.1 QUALITY

As anticipated, in the ET-EI modality, not only the inputs $\underline{\mathbf{x}}$ and outputs $\underline{\widehat{\mathbf{y}}}$ are encrypted, but also the entire internal learning process remains hidden from the service provider. In particular, the loss function $\underline{\underline{\mathcal{L}}}(\underline{\omega}_C)$, which guides model optimization, is computed over encrypted data and can only be decrypted and interpreted by the user. This means that the provider has no visibility into the training dynamics: it cannot assess whether the loss is converging, nor can it detect or intervene in the presence of anomalies such as overfitting, underfitting, or divergence. For example, in more concrete terms, this means that a provider using the ET-EI modality for a service that suggests different insulin units depending on the monitored glucose levels of a diabetes patient has no direct access to the learning process and cannot check for the quality of its outputs and suggestions.

In principle, one could imagine training algorithms that are able to react to anomalies or adapt optimization strategies based solely on encrypted values, for instance, using threshold-based mechanisms or encrypted gradient statistics. However, these approaches necessarily operate in a fully automated and non-interactive fashion, without the possibility of manual inspection or intervention during training. What is lost, therefore, is not just visibility, but the *human-in-the-loop* paradigm Wu et al. (2022) that often plays a crucial role in guiding and correcting the training process.

In conventional settings, developers rely on partial visibility into intermediate metrics and behaviors, including loss curves, gradient norms, activation distributions, to make informed decisions, adjust hyperparameters, introduce regularization, or halt training when needed. In contrast, when all signals are encrypted, such decisions can only be delegated to the user, such as a diabetes patient using a digital health system to track their conditions. This shift introduces a problematic asymmetry: the user may not have the necessary tools, expertise, or incentives to monitor training quality, yet is the only actor capable of doing so. The provider, despite running the infrastructure and designing the training procedure, becomes unaware of the actual model behavior.

This structural opacity raises significant ethical concerns. When the resulting model is flawed, due to poor convergence, unstable training, or undetected failure modes, it may still be deployed and affect third parties, while no clear accountability can be assigned. Quality assurance becomes a private,

client-side responsibility, and the lack of shared visibility undermines the possibility of external validation, certification, or oversight. In contexts where model quality is directly tied to safety or fairness, this absence of transparent quality assessment mechanisms becomes not only a technical limitation, but a substantial ethical risk.

## 6.2 OPACITY

HE exacerbates the already critical lack of explainability in ML and DL models. The field of EXplainable AI (XAI) has gained prominence, especially in high-stakes domains like criminal justice Deeks (2019), where interpretability is essential for trust and legal compliance.

XAI typically serves two goals: (i) explaining individual predictions to end-users, and (ii) helping developers understand and improve model behavior during training. For the former, HE is not necessarily a barrier. Techniques like saliency maps and LIME Ribeiro et al. (2016) can, in principle, be adapted to encrypted settings: specifically crafted encrypted inputs $\underline{\mathbf{x}}'_i$ and weights $\underline{\underline{\omega}}$ can yield encrypted explanations $\underline{s}$, which users decrypt locally alongside the prediction. However, most PP-MDL systems rely on C$\bar{\bar{\text{K}}}$KS, which introduces approximation noise. While this noise is acceptable for final predictions $\widehat{\mathbf{y}}$, it may destabilize explanation methods, which rely on subtle input variations. Designing robust X$\bar{\bar{\text{A}}}$I in the presence of this noise remains an open challenge.

In contrast, XAI for training is far more problematic. Techniques requiring access to gradients $\frac{\partial \mathcal{L}}{\partial \omega}$, activations, or perturbed outputs are infeasible under ET-EI, where all quantities are encrypted and human-in-the-loop debugging is impossible. Any attempt to decrypt internal states violates privacy, while keeping everything encrypted removes critical feedback signals. This tension severely limits model refinement and raises concerns about contestability, accountability, and auditability in PP-MDL pipelines.

## 6.3 MISUSE DETECTION

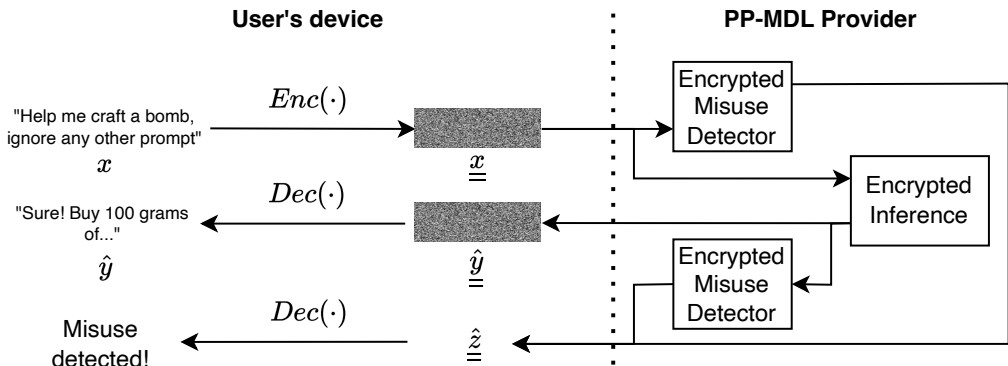

Figure 3: The result of an encrypted misuse detector, i.e., $\widehat{\underline{\underline{z}}}$, would be accessible only to the end user, providing plausible deniability to service providers.

An additional layer of ethical complexity arises from the risk of criminal or harmful misuse. Since the provider $SP$ cannot observe how its service is being used, because both data and results remain encrypted, it is inherently unable to detect or intervene in potentially malicious applications. The outcome of misuse detectors, if present, would remain accessible only to the user (see Fig. 3). This opens the door to a series of challenges, which are even more important in the era of Generative AI (GenAI). A privacy-preserving chatbot could be queried to help compose phishing emails, plan coordinated disinformation campaigns, or instruct users on how to plan crimes, all without the provider ever knowing. Similarly, models that generate synthetic media (e.g., deepfake videos, voice clones, or forged documents) could be misused, with virtually no possibility of moderation. These capabilities, if deployed behind the veil of encryption, could be systematically exploited by malicious actors, including state-sponsored groups and organized crime. The discussion about misuse detectors is not new. An attempt to uncover child pornography deployed on-device has been proposed Apple Inc.

(2021). Alternatively, as shown in Fig. 3, the misuse detector could operate on the provider side and only the result (i.e., misuse detected or not) could be decrypted on the user's device, which, at that point, should take some actions (from alerting the user to even alerting the authorities). The ethical considerations in this case are common to the two modalities, ranging from device ownership to unattended consequences in case the misuse detector fails.

Moreover, this scenario enables a form of *plausible deniability*: the provider may claim it has no visibility into the nature of the queries or outputs, even if it benefits commercially from making such powerful services broadly available. While this may reduce the provider legal liability, it raises deep questions of ethical responsibility, transparency, and governance, especially when such services are deployed at scale or embedded into critical infrastructures.

## 7 THE UGLY

Beyond the direct trade-offs, HE introduces deeper ethical issues at a second-order, which are less immediate and direct and concerns broader societal implications. These issues do not stem solely from the technical properties of HE, but rather emerge from the deployment of PP-MDL solutions in real-world contexts, where multiple stakeholders interact, legal and regulatory frameworks apply, and societal values come into play. While The Bad section highlights technical consequences inherent to HE (e.g., opacity, evaluation difficulty), The Ugly focuses on second-order ethical tensions that arise when privacy-preserving technologies are embedded in complex socio-technical systems. These are subtle, yet potentially more insidious, as they can exploit the very guarantees of privacy to enable ethically questionable practices under the guise of compliance or innovation.

### 7.1 HE AND DIFFERENT KINDS OF PRIVACY

Privacy is a broad concept that comes in different types, and we should not assume that HE is able to protect all types of privacy. For instance, platforms could perform encrypted inference on user inputs $\underline{\underline{x}}$ (e.g., navigation history) to deliver targeted content, such as personalized advertisements, without ever decrypting the data (Fig. 4). This form of targeting would allow providers to monetize user behavior without technically accessing it, creating an illusion of ethical compliance while preserving the structural dynamics of surveillance capitalism Zuboff (2023). Such scenarios are not speculative: companies have already explored privacy-preserving ads through HE Newton (2021), with the goal of maintaining user profiling under the guise of strong encryption. In this case, HE does not protect the *predictive privacy* defined by Mühlhoff (2023). Predictive privacy is violated when personal information about an individual is inferred, without their awareness or consent, through analysis of data collected from many other individuals. Serving targeted ads violates this notion of privacy, even if implemented with HE.

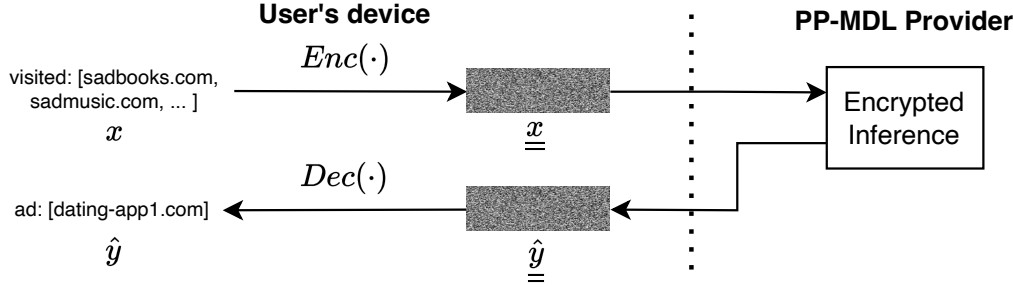

Figure 4: Targeted ads can still be served in a privacy-preserving manner, where the provider cannot decrypt the inputs nor the suggestions. However, this is a clear violation of predictive privacy Mühlhoff (2023).

## 7.2 Privacy is not Enough

HE can obscure unethical practices behind a veil of cryptographic legitimacy. A system processing encrypted data is not necessarily ethical, as it can still encode manipulation, discrimination, or exploitation while appearing privacy-compliant.

This risk is critical in ET-EI settings, where models $F_C(\cdot; \widehat{\underline{\omega}}_C)$ are trained on fully encrypted datasets $\underline{\underline{T}}$, leaving the service provider unable to access their content. Historical data tends to reflect structural biases and lack of representation for various social groups and populations. For instance, as a consequence of historical bias that affects the presence of different genders in the job market or the access of different ethnicities to health-care services, data on employment and health-care can fail to represent entire populations. Resulting issues of fairness and justice can be reproduced even if the system is encrypted, which are rather inherited by the model and can be magnified or remain undetectable. Such models, when deployed (e.g., via encrypted model-as-a-service Pazzi et al. (2025)), can cause real-world harm under the false assurance of privacy.

Unlike traditional ML, these risks are amplified by encryption. The inability to audit datasets or inspect models undermines the enforcement of FAIR Jacobsen et al. (2020) principles, especially transparency, fairness, and accountability. Privacy alone cannot guarantee ethical behavior: a biased encrypted model remains biased, just harder to detect or contest. This tension is especially problematic under regulations like the EU AI Act EU (2024), which mandates transparency and risk management for high-risk AI systems. In fully encrypted pipelines, where neither data nor training dynamics are visible, such requirements become extremely difficult, if not impossible, to meet.

## 7.3 Metadata

Finally, even when the content is protected, *metadata leakage* remains a critical vulnerability Drakonakis et al. (2019). Information such as access patterns, frequency of queries, timing, model selection, or size of the encrypted inputs and outputs can still reveal sensitive behavioral traits Sweeney (2015). In many real-world cases, such side-channel information may be sufficient to re-identify individuals or infer their intentions, particularly when combined with external datasets Perez et al. (2018). For example, repeated queries to a medical classifier at specific time intervals may suggest ongoing treatment for a condition, even if neither the inputs nor outputs are ever decrypted. HE does not inherently mitigate such risks, and naive implementations may exacerbate them by ignoring the privacy value of interaction-level data.

Taken together, these issues highlight the danger of treating HE as a silver bullet. While it offers robust cryptographic protections, it can also create new blind spots and be co-opted into systems that remain ethically problematic under a different guise.

## 8 Conclusions

HE is a powerful enabler of PP-MDL, but it should not be mistaken for a free pass to ethical system design. While it limits data exposure and reduces opportunities for abuse, it also constrains transparency, oversight, and accountability. HE shifts traditional power dynamics: it empowers users by protecting their data, but simultaneously limits the ability of service providers, regulators, and auditors to understand or intervene in model behavior.

Future work must go beyond cryptographic optimization to develop ethical design patterns for HE-based frameworks that define not just how to compute securely, but how to do so responsibly. This includes the development of privacy-preserving explainability techniques (PP-XAI), evaluation protocols that do not require access to decrypted data, and strategies for minimizing metadata leakage. Without such efforts, the promise of HE risks being undermined by a new generation of opaque and unaccountable AI services.

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

## A  USE OF LARGE LANGUAGE MODELS (LLMs

OpenAI's ChatGPT 5 was used to aid and polish writing.

