# OpenReview forum: "The Ethics of Privacy-Preserving Deep Learning: the Good, the Bad, and the Ugly"
_ICLR.cc/2026/Conference — ICLR 2026 Conference Withdrawn Submission_

### Official Review · Reviewer_uGRo · 2025-10-24

**Soundness:** 2
**Presentation:** 3
**Contribution:** 1
**Rating:** 2
**Confidence:** 4

**Summary:**

This paper considers the ethical implications of using Homomorphic Encryption (HE). Specifically, they investigate the desirable outcomes (The Good), trade-offs related to accountability, interpretability and responsibility (The Bad) and ways in which HE can be used to mask unethical practices (The Ugly). They highlight the practical limitations of using HE that limit visibility into model training and make it harder to explain predictions. Additionally, they argue that HE  does not eliminate side-channels which can be used to compromise user privacy.

**Strengths:**

The paper does a good job of explicitly listing all the practical constraints and ethical concerns of using HE to train and infer on private data.

**Weaknesses:**

1. “This shift introduces a problematic asymmetry: the user may not have the necessary tools, expertise, or incentives to monitor training quality, yet is the only actor capable of doing so. The provider, despite running the infrastructure and designing the
training procedure, becomes unaware of the actual model behavior.”

- Seems like this concern would arise with any privacy preserving training technique, not just HE. So to some extent, this issue seems fundamental to the task of privacy preserving training. This  needs to be better explained in the paper.
- Also, why is this an ethics problem? Seems more related to practical constraints of privacy preserving training.

2. The paper proposes no solutions (beyond general advice) to any of the limitations that they point out with HE.

3. I did not find the claims of this paper to be novel. Most of the claims are related to highlighting practical tradeoffs of using privacy preserving training/inference (e.g. we can’t get additional information that could be useful for monitoring training, doing explainable ai etc). Generally, I think it is well understood that privacy comes at a cost. This paper simply enumerates all the tradeoffs one has to make while adopting privacy preserving training/inference.

4. Misuse detection: Wouldn’t it be possible for a service provider to gate the response of the model based on the response of the misuse detector? In figure 3, instead of sending two separate responses for the encrypted inference and misuse detector, you could send a single gated response (if misuse_detected send dummy_response, else send original_response).

**Questions:**

Is there a reason why the concerns in the paper are attributed to HE? Seems like most of the concerns are generally applicable to any privacy preserving technology.

---

### Official Review · Reviewer_eYeV · 2025-10-28

**Soundness:** 2
**Presentation:** 3
**Contribution:** 1
**Rating:** 2
**Confidence:** 4

**Summary:**

This paper explores the ethical dimensions of using Homomorphic Encryption (HE) in privacy-preserving machine and deep learning (PP-MDL). It organizes the discussion into three perspectives: The Good, The Bad, and The Ugly, highlighting privacy benefits, ethical trade-offs, and potential misuse scenarios.

**Strengths:**

1. The paper raises an underexplored and important question: what are the ethical consequences of HE in AI systems?
2. The “Good, Bad, Ugly” structure provides a clear narrative flow and helps organize complex ethical considerations.

**Weaknesses:**

1. There is no empirical validation or case studies to demonstrate the proposed ethical risks in real or hypothetical systems.
2. Most problems and notations are descriptive, not analytic. The whole work lacks theoretical depth.

**Questions:**

1. Is that possible to add some empirical validation or case studies?
2. Or any actionable mechanisms?

---

### Official Review · Reviewer_1BKR · 2025-10-30

**Soundness:** 3
**Presentation:** 4
**Contribution:** 2
**Rating:** 4
**Confidence:** 3

**Summary:**

This paper discusses the positive and negative ethical implications of homomorphic encryption (HE) for privacy-preserving ML and deep learning (PP-MDL); particularly, the goal is to explore the ethics of HE beyond plain privacy. The main message is the HE does not automatically solve all ethical issues of PP-MDL, and it incurs inherent tradeoffs between cryptographical privacy and practical ethical concerns.

**Setting**: The paper considers two settings: i) "Plain" training and encrypted inference, where a model is trained on public data, and a user encrypts their inputs and obtains an encrypted inference result that only they can decrypt. ii) Encrypted training+inference, where the entire training data is encrypted with a single key, and the training procedure uses homomorphic encryption, yielding encrypted weights. For the sake of discussion, the paper assumes that both settings are practically feasible.

**Discussion**: The paper groups its discussion into three categories: What are ethically desirable outcomes of HE ("good"), what are direct technical tradeoffs from using HE ("bad"), and what are indirect, higher-order ethical issues that result from deploying PP-DML in practice ("ugly").

**Good (ethically desirable outcomes of HE in PP-MDL)**:
- NB: In this context, the paper argues that preserving privacy implies being ethical.
- HE can, under the assumptions of this paper, be applied to existing AI services to make them private. This enables inference for data which is protected (e.g., medical data) or could more generally violate contextual integrity (e.g., satellite images in a military context).
- HE is also a building block to improve other privacy-enhancing technologies (e.g., federated learning or private information retrieval).

**Bad (direct technical tradeoffs from using HE)**:
- For HE training, a model provider cannot observe and act on the training process (e.g., does the loss converge) or evaluate the trained model. This shifts the burden to the end-user, who holds the only key.
- HE inference and training introduces challenges to model explainability due to numerical issues or lack of (unencrypted) training statistics.
- HE hinders misuse detection. For HE training, a model provider cannot reject unethical datasets. For HE inference, a model provider might be able to prevent unethical inference, but cannot learn that an inference request was refused.

**Ugly (indirect ethical issues from deploying PP-MDL in practice)**:
- HE does not preserve predictive privacy. That is, it still allows inferring personal information about an individual (e.g., serving targeted ads) and does hence not prevent unethical practices.
- A model trained with HE might still contain unethical biases (e.g., being unfair). This is obfuscated via encryption, and regulatory auditing is impossible on encrypted weights.
- Even if training and inference are fully encrypted, metadata (e.g., access patterns) might act as a side-channel that enables privacy inference.

**Strengths:**

**Novelty**: While the ethical tradeoffs of cryptographic privacy have been discussed in general, this paper explicitly considers the ethics of HE in the context of machine learning and raises important points. This particular combination of HE and PP-MDL has, to the best of my knowledge, so far been overlooked in the literature. It is hence crucial to discuss the ethics of HE in deep learning before it becomes practically feasible and widely adopted.

**Argumentation**: The discussion points are stated in a clear way, and the authors provide intuitive examples to support their arguments. I do agree with most high-level points made in this paper.

**Presentation**: The authors provide a clear and self-contained introduction to HE that is sufficient for readers unfamiliar with HE. I also like that the authors try to clearly specify two settings that they use as the base for discussion. The paper is overall written well.

**Weaknesses:**

**Significance of HE for training**: It would be great if the paper could stress the significance of HE training more, because it is referenced in many arguments. First, HE itself does not enable collaborative training (such as e.g. the one mentioned on L261-266). Even naive secure multi-party computation does not preserve privacy, as the resulting model is susceptible to privacy attacks. Hence, collaborative training would require a mix of e.g. HE, federated learning, and differential privacy, which is out of the scope of this paper.

In addition, cannot think of many use cases that require both i) the strong privacy of HE (with its computational drawbacks) and ii) rely on a single user/entity's data. Motivating HE training would hence strengthen the overall arguments.

Lastly, while the rest of Section 4 is clear, the setup of HE training+inference in Section 4.2 excludes the inference part. However, this is significant missing detail, because it determines which applications are possible.

**Relatively narrow topic**: The scope of this paper is explicitly *homomorphic encryption* for *privacy-preserving ML/deep learning*. While the arguments are tailored to this niche, and there is merit in discussing them, I am not sure how big this paper's practical impact is. In particular, developing viable HE for PP-MDL (esp. training) could require years of research. During that time, both the regulatory landscape and public opinion might change. Hence, the paper could benefit from stressing its importance *today* more, e.g., by mentioning actionable insights that should be taken now.

**Minor points**
1. The main argument in Section 6.1 is that a model trainer cannot intervene in HE training, shifting all the burden to the user. However, this might be partially solvable (e.g., ZKP for loss convergence). This does not fully invalidate the argument (e.g., human-in-the-loop paradigms are still challenging) but weakens it.
2. This paper's goal is to explore ethics beyond the benefits of cryptographic privacy. However, Section 5 only considers privacy benefits, while Sections 6.1 and 6.2 only consider purely technical issues. A different framing could strengthen the paper's arguments.

**Questions:**

Who exactly can perform inference in the "encrypted training and inference" setting (Section 4.2)? Is it only the key used for data encryption (Equation 8)? In general, what is the "threat model" considered there?

---

### Official Review · Reviewer_Px24 · 2025-11-02

**Soundness:** 2
**Presentation:** 2
**Contribution:** 1
**Rating:** 2
**Confidence:** 3

**Summary:**

This paper explores the ethical implications of using Homomorphic Encryption (HE) for Privacy-Preserving Machine and Deep Learning (PP-MDL). The authors structure their analysis around three perspectives: "The Good" (privacy benefits), "The Bad" (direct ethical trade-offs around accountability and transparency), and "The Ugly" (second-order societal implications). The paper formalizes PP-MDL services under HE in two modalities: Plain Training-Encrypted Inference (PT-EI) and Encrypted Training-Encrypted Inference (ET-EI), then discusses how HE affects various ethical dimensions including quality assurance, explainability, misuse detection, and broader privacy considerations.

**Strengths:**

1. Timely systemization: The paper provides a useful systematization of knowledge around the ethical implications of HE in AI systems. Given the increasing interest in privacy-preserving ML, having a structured discussion of trade-offs is valuable.

2. Balanced perspective: I appreciate that the authors discuss both benefits and drawbacks together, moving beyond the typical "privacy-as-panacea" narrative that dominates much of the HE literature.

3. Breadth of ethical dimensions: The paper touches on multiple important ethical values beyond privacy (accountability, transparency, quality, fairness, etc.), which is important for a holistic ethical analysis.

**Weaknesses:**

1. Limited technical novelty and unclear venue fit: This paper reads primarily as a position paper or systematization of knowledge rather than a technical contribution. While such papers have value, ICLR typically emphasizes novel technical contributions, methods, or empirical insights. The paper contains no new algorithms, experimental results, or formal analyses. I'm concerned this may not be the right venue - a workshop, ethics-focused venue, or journal might be more appropriate.

2. Missing critical related work and incomplete literature coverage: The paper has significant gaps in its treatment of related privacy-preserving techniques and their ethical implications:

3. No substantive discussion of Differential Privacy (DP): DP is conspicuously absent despite being a major privacy-preserving approach with its own well-studied ethical trade-offs. The paper should discuss how HE's ethical landscape compares to and differs from DP. Relevant missing work includes:

Priyanshu A, Naidu R, Kumar A, Kotti S, Wang H, Mireshghallah F. "When Differential Privacy Meets Interpretability: A Case Study."
Papernot N, Steinke T. "Hyperparameter Tuning with Renyi Differential Privacy." ICLR.


4. Incomplete treatment of privacy risks: Section 7.3 briefly mentions metadata leakage but doesn't engage with the substantial literature on privacy risks of model explanations:

Shokri R, Strobel M, Zick Y. "On the privacy risks of model explanations." AAAI/ACM Conference on AI, Ethics, and Society, 2021.


5. Insufficient engagement with contextual integrity: While the paper nicely invokes Nissenbaum's contextual integrity theory in Section 5, the treatment is superficial. More problematically, recent work has actually applied contextual integrity theory to analyze privacy in ML systems, which is highly relevant but not cited:

Mireshghallah N, Kim H, Zhou X, Tsvetkov Y, Sap M, Shokri R, Choi Y. "Can LLMs Keep a Secret? Testing Privacy Implications of Language Models via Contextual Integrity Theory." ICLR 2024.


6. Conceptual confusion about contextual integrity: The paper claims HE supports contextual integrity (Section 5, lines 269-277), but this needs more careful analysis. Contextual integrity is about appropriate information flows within specific social contexts - it's fundamentally about norms and context, not just technical protection. HE provides blanket encryption regardless of context, which is actually quite different from contextual integrity's framework. The paper needs to either develop this connection more carefully or acknowledge that HE provides a different kind of privacy guarantee.

7. Lack of concrete paths forward: The paper identifies many problems but offers little guidance on solutions. Section 8's conclusions mention "ethical design patterns," "PP-XAI," and "evaluation protocols" but provides no specifics. Given the substantial challenges identified in Sections 6-7, readers need more concrete proposals or at least a research agenda with specific technical directions.

**Questions:**

1. Comparison with DP: How do the ethical trade-offs of HE compare to those of differential privacy? DP also has well-known trade-offs between privacy and utility/accuracy, and between privacy and fairness. A comparative analysis would significantly strengthen the paper and help readers understand when HE vs. DP might be more ethically appropriate.

2. Paths forward: What do the authors believe is a realistic way to move forward on these challenges? Are there specific technical research directions that could address some of the "Bad" issues (like quality assurance under encryption)? Or do you believe some of these are fundamental limitations that require organizational/regulatory solutions rather than technical ones?

---

### Note · Authors · 2025-11-12

I have read and agree with the venue's withdrawal policy on behalf of myself and my co-authors.